# Prevalence of *Blastocystis* in Geese Reproductive Flocks

**DOI:** 10.3390/ani12030291

**Published:** 2022-01-25

**Authors:** Piotr Falkowski, Andrzej Gaweł, Kamila Bobrek

**Affiliations:** Department of Epizootiology and Clinic of Bird and Exotic Animals, Wroclaw University of Environmental and Life Sciences, 50-375 Wrocław, Poland; piotr.falkowski@upwr.edu.pl (P.F.); andrzej.gawel@upwr.edu.pl (A.G.)

**Keywords:** *Blastocystis*, geese, prevalence, zoonosis

## Abstract

**Simple Summary:**

Some diseases may be transmitted from animals to humans, and to prevent this, more studies on controlling infectious agents in animals are needed. We conducted an investigation among reproductive geese flocks to determine the presence of *Blastocystis*—a protozoan that may infect people and animals. The parasite was present in 46.5% of tested flocks, and there was no correlation between the bird’s age and flock size. Our conclusion is that geese could be the source of infections in humans who have contact with infected birds.

**Abstract:**

*Blastocystis* is a unicellular, anaerobic protozoan that has a low specificity for the hosts, and it could be a zoonosis. There are not many data about the occurrence of *Blastocystis* in bird species, and this study aimed to check the prevalence of *Blastocystis* infection in reproductive geese flocks. The result obtained showed that a parasite was present in 46.5% of tested flocks. The extensiveness of the *Blastocystis* invasion in reproductive geese flocks was low because the genetic material of parasites was found only in 7.48% of samples. There was no correlation between the infection and the bird’s age or the flock size. The data obtained showed that geese could be the source of infections in humans who have contact with carriers of the infection.

## 1. Introduction

*Blastocystis.* is commonly located in the gastrointestinal tract of mammals (including humans) and in birds, reptiles, and amphibians [1]. The parasite is a unicellular, anaerobic protozoan that has a low specificity for the hosts—it is transmitted between various species [2,3], mainly via the fecal–oral route due to polluted food and water [4]. *Blastocystis* varies extensively in both shape and size, with a 3 μm to 120 μm range in diameter and with four major morphological forms described [5,6,7,8]. The *Blastocystis* infection symptoms in humans rarely occur. In a divergent manner, they include acute or chronic diarrhea, abdominal pain, nausea, anorexia, bloating, fatigue, and flatulence or urticaria [9,10]. The controversy around the clinical potential of *Blastocystis* is grounded on a high rate of asymptomatic carriers and no certainty of whether the clinical symptoms of *Blastocystis* infection are related to a determined subtype, a number of subtypes, or colonization by a multitude of parasites [11]. There has been speculation recently that parasites have a positive effect on the intestine microbiota and ought to be deemed commensal [12]. However, this parasite affects human health, and its exact influence on the human condition is still unknown [13]. In vitro studies proved the presence of some virulence factors, including cysteine proteases and mechanisms, which may take part in the pathogenesis of the parasite [14]. Additionally, some studies demonstrated the occurrence of the parasite in the digestive tract stimulating the development of *Campylobacter*, as demonstrated in broiler chickens [3]. Numerous aspects of *Blastocystis* biology, epidemiology, and pathogenicity remain unresolved, despite its frequent occurrence in humans and animals worldwide. This parasite has been globally recognized in mammals and birds, which renders it a potential source of human infection [15]. It has been confirmed that *Blastocystis* is carried by wild and domestic birds. Domestic poultry is a potential source of zoonotic parasite transmission since it breeds in close proximity to humans. Its feces are often used as fertilizer for crops cultivated for human consumption. A significant variation in parasite prevalence in chickens was shown in past surveys [16]. Among 17 subtypes of *Blastocystis*, in chickens, subtypes 6 and 7 were noted. The slaughterhouse staff were infected with subtype 6, when most of the people in the region were carriers of subtype 3. This confirms the zoonotic transmission of *Blastocystis* [3]. The most prevalent subtypes in Polish people are 1–3 and 7, with a low (6–10%) prevalence. European people are generally infected by *Blastocystis* subtypes 1–4 [17,18]. In Polish chickens, subtypes 5–7 were noted [19]. However, the data about the prevalence of *Blastocystis* in bird species other than chickens are rare. Poland, an important producer of geese for the European market, produces over 50,000 tonnes of geese meat annually.

Considering the lack of current data related to the occurrence of *Blastocystis* in geese and its zoonotic potential, this study aimed to define the level of occurrence for this protozoan in reproductive geese flocks, taking into account the birds’ ages and the flock size.

## 2. Materials and Methods

### 2.1. Sample Collection

This investigation was conducted over four years (2016–2019) in five provinces of Poland with the highest number of parental geese flocks. The 43 reproductive geese flocks, which were kept in separate farms, were examined. The samples were taken after laying cycles from flocks defined as healthy (with no symptoms of disease). Cloacal swabs were taken from 23 geese randomly chosen from each flock, according to the Biocheck manual for poultry flocks sampling [20]. They were then placed separately.

Each flock has been classified according to the number of past production cycles (first, second, and third) and the flock size (small flocks had less than 1000 birds, medium flocks had 1001–2000 birds, and large flocks had over 2000 birds). The birds were tested only once during the entire production cycle.

### 2.2. Parasite Culture

Each cloacal swab was placed in liquid Medium 199 with HEPES, and 15% serum of newborn calves was added immediately after swabbing. The swabs were incubated at 37 °C for 48 h with monitoring after 24 h. After 48 h, a 2 mL culture suspension was taken and used for DNA isolation [21].

### 2.3. Genetic Material Extraction and PCR

The DNA was isolated from the *Blastocystis* culture, which was centrifuged in 400× *g* 5 min. The genetic material was extracted from the sediment using a commercial Genomic Mini kit (A&A Biotechnology) in accordance with the manufacturer’s instructions. The samples were stored at –20 °C for future usage.

PCR was carried out following the method described by Grabensteiner and Hess [22] at the following cycling conditions: initial denaturation at 94 °C for 15 min followed by 40 cycles of 94 °C for 30 s, 60 °C for 30 s, 72 °C for 1 min, and a final extension at 72 °C for 10 min. The amplification products were resolved in a 2% agarose gel, stained with Midori Green, and visualized using Gel-Doc UV transilluminator system with Quantity One software (Biorad). The products (size ~500 bp) from 5 cultures were sequenced to confirm the genus *Blastocystis* belonging. The obtained sequences were analyzed using Mega X and compared with GenBank sequences. One sample from the parasite culture was chosen as a positive control for future investigation, and the sequence obtained was placed in the GenBank database under accession number MZ956599.

### 2.4. Statistical Analysis

For statistical analysis, a mixed logistic regression model was used. As the flock was a surveyed unit with grouped birds, they were adopted as a random effect. To confirm the significance of these results, post hoc tests were performed using simultaneous tests for general linear hypotheses. The *p*-value of 0.05 was taken as the limit for significance with 95% confidence intervals. Calculations were made in the software R, version 3.6.0. The flock was considered infected if at least one sample among 23 was positive. 

## 3. Results and Discussion

The prevalence of *Blastocystis* from 43 parental geese flocks after the first, second, and third laying cycle is shown in Table 1.

The investigation showed that the extensiveness of the parasite invasion in reproductive geese flocks was low, and the genetic material of parasites was found only in 74/989 (7.48%) of samples. The correlation between the age and flock size, with the frequency of occurrence of *Blastocystis*, is shown in Figure 1.

The percentage of infected geese of small parental flocks after the first year of production was 1.45%, which increased with age. In medium flocks, 6.21% of samples, and in large flocks, 6.52% of samples, showed the presence of parasite DNA. Positive percentages of birds after the second year of laying were as follows: for small flocks, 8.7%; for medium flocks, 2.17%; and for large flocks, 8.7%. The flocks after the third production cycle were characterized by higher numbers of infected birds. In small flocks, infected birds constituted 5.8%; in medium flocks, they constituted 15.22%; and in large flocks, they constituted 12.32%. The correlation between the percentage of positive samples, the birds’ ages, and the flock size was modeled using a mixed logistic regression model. Since the flock was used as a unit of study and a group of birds, they were taken as a random effect. Looking at the *p*-values of the fixed effects, it stated that both the age of the birds (Chisq = 2.646, df = 2, *p* = 0.2663) and the size (Chisq = 0.9102, df = 2, *p* = 0.6344) did not statistically significantly affect the percentage of infected geese. The significance of the effect of the interaction between age and flock size was also checked, and it proved not to be statistically significant (Chisq = 2.9819, df = 4, *p* = 0.5609). There was no need for post hoc analysis.

However, an increase in the total percentage of infection with the age of the flocks was noted. The overall distribution of infected birds depending on age was as follows: 5.48% of positive geese after the first laying cycle, 7.07% of positive birds after the second laying season, and 11.07% of positive birds after the third production cycle. The parasite was detected in 20/43 (46.5%) of examined flocks.

Additionally, a comparison between 24 h cultured incubation microscopic examination and PCR examination was made. A microscopic evaluation proved successful in a total of 97.3%.

*Blastocystis* as a part of the human microbiota, commensal, or pathogen, is still a relatively unknown parasite, which cycles in the human and animal environment. The data about *Blastocystis* occurrence in poultry flocks are scarce, and the following report examines it in geese flock. A study from Lebanon reported that a major presence of *Blastocystis* (31.8%) was observed in three large poultry slaughterhouses, as shown in a molecular study conducted on 223 bird specimens collected [3]. A comparable occurrence was observed in chickens in Indonesia (34.2%), Ivory Coast (28.6%), and Libya (33.3%) [23,24]. Four studies used microscopy for diagnosis and reported a high prevalence of *Blastocystis*: 74.5% (169/227) in Australia, 32.9% (23/70) in Brazil, 29.4% (50/170) in India, and 25.2% (27/107) in Malaysia [25,26,27,28]. A low *Blastocystis* prevalence ranging from 4 to 13% in chickens was determined in three studies [29,30,31,32]. Some observations of non-chicken poultry species proved a particularly high occurrence in domestic ducks (80% infection) [33] and ostriches (100% infection) [34]. When compared with the occurrence of over 30% *Blastocystis* presence in chickens, our results are convergent but lower than in ducks. The study confirms that *Blastocystis* is frequently found in the reproductive flocks of geese in Poland. Birds with *Blastocystis* infection were 7.82% of the sampled flocks. Water was the source of infection; that is, infective water-resistant cysts stayed alive in puddles and drinkers. *Blastocystis* is reported to thrive in various water resources despite using water treatment technologies. As geese reproductive flocks have contact with the external environment, it may be speculated that birds might become infected by interaction with mammals and birds in the wilderness [35]. The low occurrence in the flock shows that the infection has a self-limiting character with an impact of the immune system. The mice-model experiment proved that the infected immunosuppressed mice were affected to a larger extent than immunocompetent mice with the infection [36], and geese may be similar in this aspect. *Blastocystis* in humans can colonize numerous individuals; however, the onset of the infection depends on the relation between the parasites’ virulence and the effectiveness of the host’s immune system [37]. Gastrointestinal tract parasitoses has a crucial role in the morbidity and mortality of patients with acquired immunodeficiency syndrome (AIDS) and human immunodeficiency virus (HIV) [38]. Humans with lowered immunological system effectiveness and diseases of the gastrointestinal tract such as colorectal cancer are more susceptible to infections [39,40]. As the evidence shows, *Blastocystis* sp. has an important role in carcinogenesis enhancement by means of damaging the intestinal epithelium [41]. Due to the not yet fully known pathogenesis of many intestinal diseases in humans and animals and the possibly zoonotic nature of *Blastocystis* infections, the need for epidemiology investigations of *Blastocystis* infections in animals is important. 

Our research showed that the size of the flock and the birds’ ages had no impact on the *Blastocystis* prevalence. It was opposite to the results obtained during the *Tetratrichomonas* prevalence research, in which older birds had higher parasite infestations [42]. Lee and Stenzel [25] investigated a property and the conditions of its environment, concluding that properly cleaned floors, utensils, and equipment; the removal of feces; and clean food and water containers decreased the possibility of infection. We suspect that proper sanitary conditions could be possible inhibitors of environmental contamination, preventing poultry infections.

## 4. Conclusions

Geese could be a potential source of *Blastocystis* in humans and other animals. The parasite was present in 46.5% of tested flocks, but the extensiveness of the invasion in flocks was low (7.48%). To our knowledge, the paper presents the first research indicating that *Blastocystis* is widespread in geese reproduction flocks.

## Figures and Tables

**Figure 1 animals-12-00291-f001:**
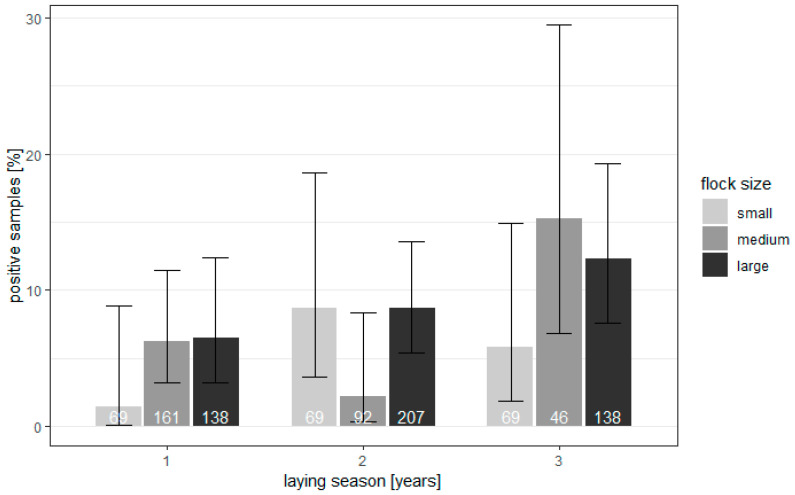
Flock and bird level prevalence of *Blastocystis* in geese from 43 flocks in Poland (95% confidence intervals).

**Table 1 animals-12-00291-t001:** The prevalence of *Blastocystis* infection in reproductive geese flock. Twenty-three samples from each flock were taken.

Laying Season (After)	1st	2nd	3rd	Total
Number of examined flocks	16	16	11	43
Number of infected flocks	7	7	6	20
Percentage of infected flocks (95% CI) [%]	37.5 (20.8–69.4)	43.75 (20.8–69.4)	63.7 (24.6–81.9)	46.5 (31.5–62.2)
Samples positive/taken	20/368	26/368	28/253	74/989
Mean percentage of infected birds (95% CI) [%]	5.48 (3.43–8.40)	7.07 (4.76–10.3)	11.07 (7.60–15.8)	7.48 (5.96–9.35)

## Data Availability

Not applicable.

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
