# Peer review of "Prevalence of Blastocystis in Geese Reproductive Flocks"

_animals, 2022, doi:10.3390/ani12030291_

Round 1

Reviewer 1 Report

The manuscript reports on a cross-sectional study estimating flock and individual prevalence of Blastocystis spp. in geese and potential association with age and flock size. The study appears well conducted but has several aims and adequate reporting may need the paper to be changed into a full article; a communication format may not allow enough space to provide this information, necessary for assessing the research's internal and external validities.    

Note to the editor: I comment as a study design and observational studies expert, but I do not have subject matter knowledge regarding blastocystis and geese farming. Please make sure these are covered by another reviewer if possible at all. 

Introduction
Please indicate which subtypes have been found in poultry and which have been suspected as having transmission potential between poultry and humans from multispecies studies. 
For context please also provide some information on prevalence in humans in the study country or region, and how important is geese farming. As it is the introduction does not make it clear why these species should be investigated
I would suggest to reword the aims to include estimation of bird and flock level prevalence as well as age and flock size effects. 

Material and methods
Please indicate any animal ethics consideration/approval number. 
Please add your sample size calculation. 
Please add in the methods your definition of an infected flock as samples were collected at animal level, as well as statistical tests conducted. 
Please add the sensitivity and specificity of the diagnostics procedure and calculate adjusted flock prevalence. 

Results and discussion
Please make sure the tables and figures are appropriately captioned. The caption should be explicit without having to read the text; for example the caption for Table 1 could be "Flock and bird level prevalence of Blastocystis spp. in geese from 43 flocks in Poland"
Please present the full results of the logistic regression model, for example in a table showing the coefficients and SE or better OR and confidence intervals. This will also help with interpretation as the sample size is quite low (43 flocks, considering your dependent variables are all flock-level). 
l 79-81 and Table 2: I would suggest to delete the comparison of PCR with microscopic evaluation. This is not part of the stated objectives of the study and including it would require additional details in the methods, analyses and results.  
l 98-100 "A curious observation was made in the flock with subsequent laying cycles as it had an increased number of birds that were infected". If I understand it well, you say it is surprising that the prevalence increases with age. This is actually expected if geese are asymptomatic, long-duration cariers. The following sentence however, seems to be about results not presented in the manuscript; you need to either remove it or present the methods and results of water testing. 
l 108-116 are focussing on human disease and are out of scope. It would be more interesting in the discussion to explore the possible differences in prevalence (subtypes, bird age, production system...)
The discussion lacks a concluding paragraph summarising the importance of the findings for geese farming, public health, and the logical next steps.   

Author Response

Dear Reviewer

Thank you for considering our manuscript for publication and providing us with your comments. Below please find the points (in italics) and our specific answers and explanations.

Reviewer 1

The manuscript reports on a cross-sectional study estimating flock and individual prevalence of Blastocystis spp. in geese and potential association with age and flock size. The study appears well conducted but has several aims and adequate reporting may need the paper to be changed into a full article; a communication format may not allow enough space to provide this information, necessary for assessing the research's internal and external validities. Note to the editor: I comment as a study design and observational studies expert, but I do not have subject matter knowledge regarding blastocystis and geese farming. Please make sure these are covered by another reviewer if possible at all. 

Introduction

Please indicate which subtypes have been found in poultry and which have been suspected as having transmission potential between poultry and humans from multispecies studies. For context please also provide some information on prevalence in humans in the study country or region, and how important is geese farming. As it is the introduction does not make it clear why these species should be investigated

Authors: Dear Reviewer, thank you for your comments. We added the information about the subtypes which were noted in people in Europe and in Poland. We refill also the informations about the Blastocystis serotypes in Polish chickens. The changes made after your suggestions are highlighted on green colour. 

I would suggest to reword the aims to include estimation of bird and flock level prevalence as well as age and flock size effects. 

 Authors: We agree. Those aims were added.

Material and methods

Please indicate any animal ethics consideration/approval number. 

Authors: There was no need to have the Ethics approval. We added this information at the Institutional Review Board Statement.

Please add your sample size calculation. Please add in the methods your definition of an infected flock as samples were collected at animal level, as well as statistical tests conducted. Please add the sensitivity and specificity of the diagnostics procedure and calculate adjusted flock prevalence. 

Authors: Thank you for the comment. We agree that the statistical part was not described properly. We added it  in Material and Methods section and in literature part. We hope that those information will find your approval.

Results and discussion

Please make sure the tables and figures are appropriately captioned. The caption should be explicit without having to read the text; for example the caption for Table 1 could be "Flock and bird level prevalence of Blastocystis spp. in geese from 43 flocks in Poland"

Authors: Thank you for your comment. The table description was changed

Please present the full results of the logistic regression model, for example in a table showing the coefficients and SE or better OR and confidence intervals. This will also help with interpretation as the sample size is quite low (43 flocks, considering your dependent variables are all flock-level). 

Authors: Thank you for the comment. We added it  in Material and Methods section and in literature part. We decide that the text description is enough for Communication. We hope that those forms will find your approval.

l 79-81 and Table 2: I would suggest to delete the comparison of PCR with microscopic evaluation. This is not part of the stated objectives of the study and including it would require additional details in the methods, analyses and results.  

Authors: We agree that the table is not necessary. We deleted it.

l 98-100 "A curious observation was made in the flock with subsequent laying cycles as it had an increased number of birds that were infected". If I understand it well, you say it is surprising that the prevalence increases with age. This is actually expected if geese are asymptomatic, long-duration cariers. The following sentence however, seems to be about results not presented in the manuscript; you need to either remove it or present the methods and results of water testing.

Authors: Thank you for the comment. We removed the sentence. 

l 108-116 are focussing on human disease and are out of scope. It would be more interesting in the discussion to explore the possible differences in prevalence (subtypes, bird age, production system...)

The discussion lacks a concluding paragraph summarising the importance of the findings for geese farming, public health, and the logical next steps.  

Authors: Thank you for the comment. We added the thesis why there was low level of infection and added the concluding paragraph. 

Reviewer 2 Report

The Communication entitled "Prevalence of Blastocystis spp. in geese reproductive flocks". Has been carefully evaluated. Although the number of investigated flocks and individuals is quite high, the quality of presentation of the generated data is low.  The MS has a descriptive character. Major revisions are needed. Please note my comments below.

Major comments: Neither in the shown Table 1 nor in Firgure 1 S.D. or S.E.M. are provided. The authors should consider adding these information to the percentages in Tab.1 and Fig.1. Addittionally, on the top of every bar it should be written n=xx reflecting the number of tested animals.

Minor comments:

L40: Consider replacing "hosted" with "carried".

L43: Consider replacing "visible" with "significant".

L44: The phrase "few surveys" is not accurate, please consider either naming the exact number or deleting "few".

L50: Consider replacing "came" with "were generated".

L51: A dot is missing behind Table 1.

L52/L55: Consider replacing the "percentage of infected birds" with the "mean percentages of infected birds" throughout the MS.

Author Response

Dear Reviewer

Thank you for considering our manuscript for publication and providing us with your comments. Below please find the points raised by the Reviewers (in italics) and our specific answers and explanations.

The Communication entitled "Prevalence of Blastocystis spp. in geese reproductive flocks". Has been carefully evaluated. Although the number of investigated flocks and individuals is quite high, the quality of presentation of the generated data is low.  The MS has a descriptive character. Major revisions are needed. Please note my comments below.

Major comments: Neither in the shown Table 1 nor in Firgure 1 S.D. or S.E.M. are provided. The authors should consider adding these information to the percentages in Tab.1 and Fig.1. Addittionally, on the top of every bar it should be written n=xx reflecting the number of tested animals.

Authors: Thank you for the suggestions. Both Reviewers suggested the changes in the statistics part. Please find if recent form is correct. 

Minor comments:

L40: Consider replacing "hosted" with "carried".

L43: Consider replacing "visible" with "significant".

L44: The phrase "few surveys" is not accurate, please consider either naming the exact number or deleting "few".

L50: Consider replacing "came" with "were generated".

L51: A dot is missing behind Table 1.

L52/L55: Consider replacing the "percentage of infected birds" with the "mean percentages of infected birds" throughout the MS.

Authors: Thank you for the comments - all minor mistakes were corrected.

Round 2

Reviewer 2 Report

Unfortunately, this Reviewer does not see the corrections in the attached PDF which were indicated as major comments:

Major comments: Neither in the shown Table 1 nor in Figure 1 S.D. or S.E.M. are provided. The authors should consider adding this information to the percentages in Tab.1 and Fig.1. Additionally, on the top of every bar it should be written n=xx reflecting the number of tested animals.

This is still missing and the above-mentioned comments have to be addressed!

Author Response

Dear Editor, dear Reviewer 2,
Thank you for your suggestions. The light blue highlights in the manuscript indicate the corrected text according to Reviewer
2.
Reviewer 2

Comments and Suggestions for Authors

Unfortunately, this Reviewer does not see the corrections in the attached PDF which were indicated as major comments:

Major comments: Neither in the shown Table 1 nor in Figure 1 S.D. or S.E.M. are provided.
The authors should consider adding this information to the percentages in Tab.1 and Fig.1.
Additionally, on the top of every bar it should be written n=xx reflecting the number of tested
animals. This is still missing and the above-mentioned comments have to be addressed!

Authors: Thank you for the review. We consult your suggestion with the statistician, because
it was not clear for the authors. The standard error of the mean
is computed for symmetric distributions and in this case the probability distribution is not symmetric. After the consultation, it was showed that for this data, showing the 95% confidence intervals is more suitable. The CI 95% was added in the table and on the Figure. We hope that the correctcions
will find your approval.

Best regards

Kamila Bobrek

Round 3

Reviewer 2 Report

no further comments.